# Research

developmental biology, evolution, palaeontology

development, Ediacaran, Ediacara biota, evolution, regulatory genes, Metazoa

**Author for correspondence:**
Scott D. Evans
e-mail: scotte23@vt.edu

†Present address: Department of Geosciences, Virginia Tech, Blacksburg, VA 24061, USA.

# Developmental processes in Ediacara macrofossils

Scott D. Evans[1,†], Mary L. Droser[2] and Douglas H. Erwin[1]

[1]Department of Paleobiology MRC-121, National Museum of Natural History, Washington, DC 20013-7012, USA
[2]Department of Earth and Planetary Sciences, University of California, Riverside, CA 92521, USA

(iD) SDE, 0000-0001-5654-8495; MLD, 0000-0001-7112-5669; DHE, 0000-0003-2518-5614

The Ediacara Biota preserves the oldest fossil evidence of abundant, complex metazoans. Despite their significance, assigning individual taxa to specific phylogenetic groups has proved problematic. To better understand these forms, we identify developmentally controlled characters in representative taxa from the Ediacaran White Sea assemblage and compare them with the regulatory tools underlying similar traits in modern organisms. This analysis demonstrates that the genetic pathways for multicellularity, axial polarity, musculature, and a nervous system were likely present in some of these early animals. Equally meaningful is the absence of evidence for major differentiation of macroscopic body units, including distinct organs, localized sensory machinery or appendages. Together these traits help to better constrain the phylogenetic position of several key Ediacara taxa and inform our views of early metazoan evolution. An apparent lack of heads with concentrated sensory machinery or ventral nerve cords in such taxa supports the hypothesis that these evolved independently in disparate bilaterian clades.

## 1. Introduction

The fossil record of complex, macroscopic community-forming organisms, including animals, begins with the Ediacara Biota (570–539 Ma). Molecular clock estimates suggest that major metazoan phyla and their constituent clades evolved prior to this period [1–4]; however, phylogenetic affinities for most of the Ediacara Biota remain enigmatic [5]. Potential explanations for this phylogenetic uncertainty include the simplicity of early animal forms, preservational biases, and lags between character acquisition and ecological success (e.g. [3,6,7]). Many Ediacara taxa may represent stem lineages of animal phyla but their diagnostic characters either were not preserved or had not yet evolved.

Comparative developmental studies of modern organisms reveal a wealth of information regarding the underlying genetic controls responsible for specific characters (e.g. [8]). Many genes highly conserved in bilaterians are present in all animals and can be found among their closest metazoan relatives (see review in [9]). Importantly, there is a growing database of information regarding developmental characters, their phylogenetic distribution and the genetic machinery underlying their expression.

Here, we use the expression of developmentally controlled features, or their absence, to evaluate the position of select Ediacara taxa. We identify characters of these organisms controlled by conserved developmental processes and suggest genetic elements likely responsible for their expression. Based on recent work, we assume that these taxa were animals. Although alternatives have been proposed (e.g. [10,11]) and certain fossils from this biota have been identified as non-metazoans [12,13], multiple independent lines of evidence support the interpretation of key taxa as animals (e.g. [1,5,14–16]).

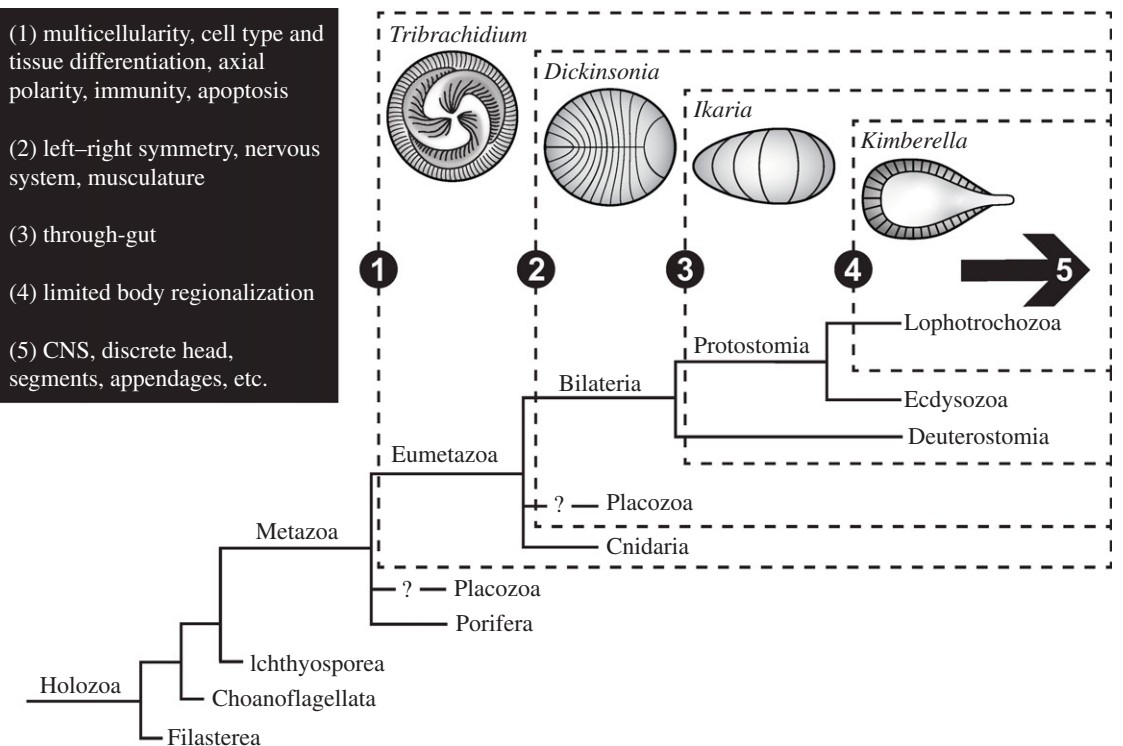

**Figure 1.** Holozoan phylogeny with inferred placement of representative White Sea taxa (dashed boxes) based on developmentally relevant characters (1–5, black box). Characters represent those that can be identified based on morphological expression in representative Ediacara fossils, and thus are not indicative of their earliest appearance. Arrow represents increased combinatorial complexity of transcription factor interactions in all three groups of bilaterians. Question marks represent uncertainty of placozoan placement. Ctenophores omitted to avoid uncertainty. CNS, central nervous system.

Observations from the fossil record rely on preserved morphological traits for recognizing potential regulatory mechanisms. Regulatory elements controlling features that are not preserved will not be recognized. Consequently, an inferred lack of a given character necessarily represents the absence of evidence.

## (a) Metazoan framework

We are concerned with the main axis of animal phylogeny, from sponges through cnidarians to the three clades of bilaterians (figure 1). Lophotrochozoans and ecdysozoans compose the protostomes, and chordates and echinoderms belong to the deuterostomes [17]. A number of problematic issues remain in metazoan phylogeny, including the position of ctenophores, placozoans and Xenacoelomorpha and the topology of major branches within the Panarthropoda and the Lophotrochozoa. The issues relevant here are the position of ctenophores, discussed below, and placozoans. In most studies, placozoans emerged after sponges and are the sister clade to all other metazoans. Recent work suggests they may be sister to cnidarians, but this result is sensitive to the position of ctenophores (e.g. [18]). We retain placozoans in their traditional position, but note that if they are sister to cnidarians they may be the remnants of a now largely missing clade of diploblastic forms, possibly including members of the Ediacara Biota.

Metazoans are classified within Holozoa, which broadly contain many of the developmental tools exploited by animals. Non-metazoan holozoans are small and have limited cellular differentiation. However, studies of major clades (filastereans, ichthyosporeans and choanoflagellates) reveal that the regulatory capacity for multicellularity, including

spatial and temporal differentiation of multiple cell types, is shared across holozoans [19,20] and thus likely present by 900 Ma [9]. A substantial increase in genome size and regulatory complexity occurred at the base of Metazoa [9,21–24]. Both morphological and more recent single cell RNA sequencing (scRNA-seq) studies have identified a dozen or more cell types in sponges, cnidarians and ctenophores (although fewer in placozoans [25,26]; summarized in [9]).

Despite expansion of the regulatory genome before and during the origin of animals, key regulatory components were not widely used until the origin of Bilateria (e.g. [27]). One example is the use of distal enhancers, regulatory sequences that lie well away from the target gene (in contrast to proximal enhancers, which are immediately upstream of their target gene). Distal enhancers are present at the base of Metazoa [28], but surprisingly, are not common in sponges, cnidarians or placozoans, possibly because more highly structured chromosome architectures were required to efficiently deploy them.

Erwin [9] proposed the following scenario: Many significant genetic processes were initially controlled by relatively flat regulatory networks largely proximal to the coding gene, limiting developmental and morphological complexity. Under this scenario, the protostome–deuterostome ancestor (PDA) was morphologically fairly simple with at most tens of different cell types. Anteroposterior (A/P) patterning was achieved largely via Wnts, while *distalless* helped generate proximo-distal patterning and Pax genes were associated with sensory activities. Near the origin of the PDA, new genes arose, the number of transcription factors (TFs) increased and regulatory potential escalated through use of distal enhancers and more structured chromosome architectures. This allowed independent co-option of conserved

genes and expansion of developmental patterning to generate complex bilaterian characters, including appendages, eyes and gut. Evidence from segmentation is consistent with this model, as it apparently arose multiple times in different bilaterian clades (e.g. [29]), and other clades, such as molluscs, that are metameric but lack true segments [30].

Although there has been considerable interest in the history of nerve cells and the early evolution of the nervous system, achieving consensus on the topic has been hampered by recent debates over the position of ctenophores. Most studies place ctenophores after sponges [31–33]. However, some analyses of molecular data support that they arose before sponges [34,35], with nervous systems evolving independently in ctenophores and eumetazoans [36,37].

Notwithstanding these issues, three primary evolutionary stages are recognized: (i) the origin of discrete neurons, likely from multifunctional sensory cells; (ii) the evolution of a diffuse nerve net; and (iii) the coalescence of a central nervous system (CNS) [38–41]. Analysis of non-metazoan holozoan clades has found evidence of proto-synaptic proteins for cell–cell communication [42] and thus, as is generally the case, many of the elements of the nervous system were present before the origin of metazoans. This facilitated the appearance of specialized neuronal cells followed by the origin of nervous systems in ctenophores, cnidarians and bilaterians (see discussion in [43]). One of the startling results from the scRNA-seq analysis is the diversity of cnidarian neuronal cell types [26].

### (b) Representative taxa

The Ediacara Biota is divided into three temporally distinct assemblages of soft-bodied, macroscopic taxa [44]. The middle White Sea assemblage is well-known from extensive deposits in Russia and South Australia [45] and both is the most diverse and has the highest morphological disparity of the three assemblages [46]. Of more than 40 recognized species, we concentrate on four representative taxa (figure 1). These exhibit features for which developmental processes are well documented among living taxa.

*Kimberella* is an approximately ovoid fossil with a broad, rounded end opposite a narrow, truncated region (figure 2*a*). The long axis can exceed 10 cm. Preservation of a significant vertical component (depth) suggests that the body was relatively thick and resilient to compaction. Association with repeated sets of scratch marks (figure 2*b*) demonstrates mobility and feeding by excavation of organic mats that lined the Ediacaran seafloor [47–50]. Morphological evidence for a projection at one end of the organism is reconstructed as a proboscis used in mat excavation [48]. The main body is interpreted with a muscular foot or analogous structure, possibly evidenced by an outer rim or 'frill' [48,49,51].

*Ikaria* are millimetre-scale, elliptical fossils (figure 2*c*) recently described from South Australia and consistent with the generation of associated horizontal burrows (figure 2*d*), *Helminthoidichnites* [52]. The preservation of negative *Helminthoidichnites* on bed soles with positive levees indicates that *Ikaria* was capable of moving through and displacing sand grains [53,54]. *Helminthoidichnites* are limited to thin (less than 15 mm) sand layers and are observed penetrating organic mats and macroscopic Ediacara taxa, evidence of scavenging [53].

*Dickinsonia* is an ovoid fossil (figure 2*e*), with one species that could grow to almost a metre in total length. It is divided down the long axis by a midline, with modular body divisions roughly perpendicular. One end of the long axis is undivided by modules. Associated trace fossils represent feeding via external ventral digestion between periods of directed, active mobility [55–58]. Claims that *Dickinsonia* may have been a giant single-celled organism [10] are contradicted by large maximum dimensions [59], mobility [55–58] and possible tissue differentiation [14,60].

The circular, triradially symmetrical *Tribrachidium* (figure 2*f*) was likely a sessile, benthic organism (although see [61]) with a maximum diameter of approximately 3 cm [62]. Threefold symmetry is rare in modern animals but is found in several White Sea taxa [63]. Results from computational fluid dynamics are consistent with suspension feeding [64].

## 2. Developmentaly controlled characters

Multicellular organisms generate multiple cell types, with tissues representing combinations of cell types, and organs spatially arranged tissues (e.g. [65]). The scale and morphological patterning of many Ediacara taxa is evidence of multiple cell types and some degree of regional differentiation [66]. Suspension feeding activity directing water to specific regions of the body in *Tribrachidium* [64] suggests the concentration of distinct cell types forming an isolated local environment consistent with tissue-grade organization [65].

Mobility in *Kimberella*, *Ikaria* and *Dickinsonia* has been attributed to muscular activity [52,55,56,67]. Muscles are composite tissues consisting of multiple cell types [65]. Feeding in *Kimberella* [48,49] and *Ikaria* [52,53] suggests the presence of a mouth and gut, potentially a through-gut, although such structures are not preserved. It is unclear whether these represent true organs, but, if present, a gut likely required multiple tissue layers, including muscles for particle transport.

*Kimberella* presents perhaps the strongest evidence for regional patterning of discrete functional units. One end of the long axis is specialized for excavation of the organic mat but remains ambiguous with respect to organ-grade differentiation. Regardless, morphological distinction, including between the surface facing upward, into the water column, and that in contact with the seafloor, represents functional regionalization.

Axial polarity and related body patterning are observed in all four taxa considered. Flipped *Tribrachidium* exhibit concentric circles on the surface in direct contact with the sediment–water interface distinct from the three-armed morphology facing upward [62]. Although polarization in *Kimberella*, *Ikaria* and *Dickinsonia* may not be homologous to A/P or dorsoventral (D/V) axes in bilaterians [15], these taxa possessed the developmental capacity to produce morphologically distinct perpendicular axes. Expression of such axes results in left–right symmetry in all three, although this may be offset in *Dickinsonia* ([57], but see [56,60]). Precise maintenance of symmetry was likely integral for functions such as mobility [55].

Despite cell-type differentiation, axial polarity and probable gastrulation, no evidence for differentiated appendages, tagmata, or sensory organs has been identified in any White Sea taxon. The absence of observable differentiation related to the long axis within repeated units in *Dickinsonia* precludes assignment as segments [68,69]. This likely extends to all contemporaneous serially divided taxa, although a segmented organism has recently been described from younger

Proc. R. Soc. B 288: 20203055

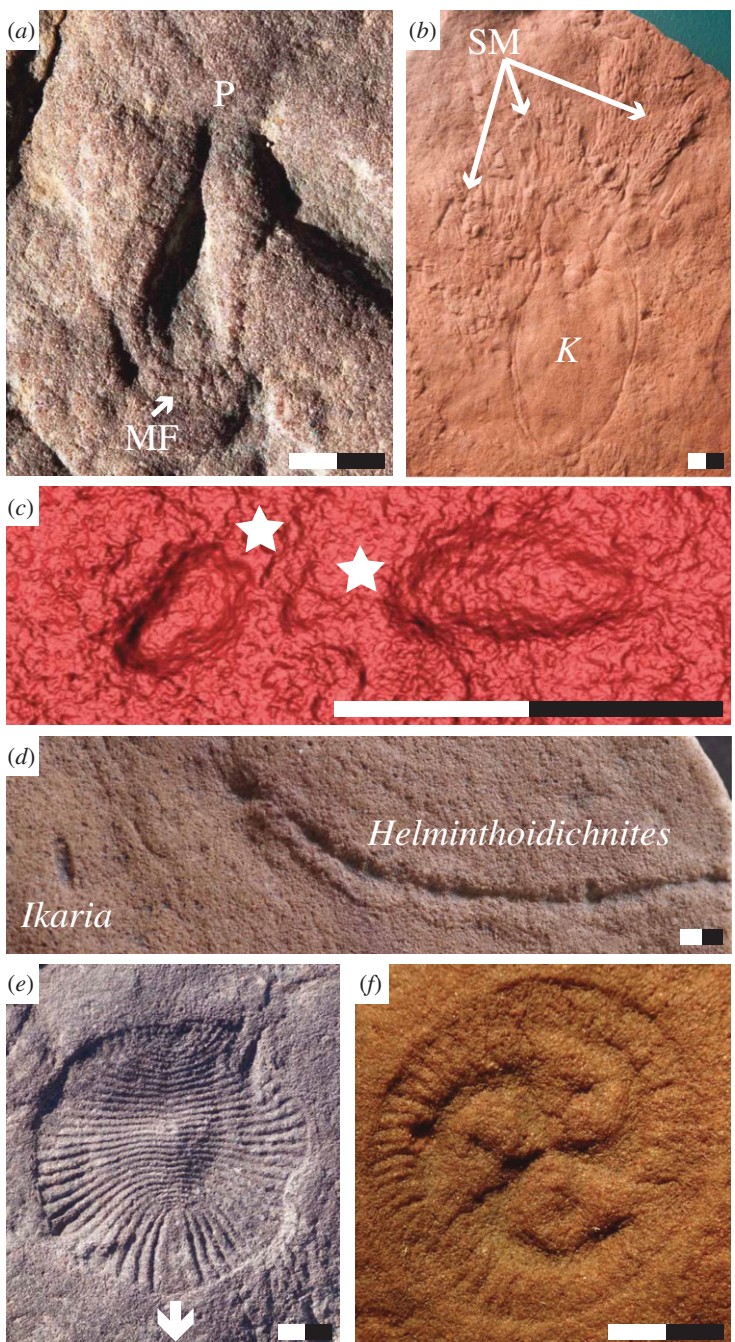

**Figure 2.** Representative taxa of the White Sea assemblage from the Ediacara Member, South Australia, including: (*a,b*) *Kimberella quadrata* (*K*) with frill or muscular foot (MF), proboscis (P) and associated scratch marks (SM); (*c,d*) *Ikaria wariootia* with wider end indicated by white stars and with associated trace fossil *Helminthoidichnites*; (*e*) *Dickinsonia costata* with white arrow indicating the direction of movement; and (*f*) *Tribrachidium heraldicum*. Fossils are external moulds preserved in negative relief on the base of fossil beds (hyporelief). (*a,b*) and (*d–f*) are photographs of original fossils and (*c*) is a three-dimensional laser scan. (*f*) Photo courtesy Christine Hall. (*a*) LV-FUN 001; (*b*) P35660; (*c*) 1T-A 001; (*d*) P57686; (*e*) TB-ARB 001; (*f*) P12898. Scale bars total 1 cm. (Online version in colour.)

Ediacaran strata in South China [70]. Representative taxa show no evidence for serial homology of repeated divisions into distinct functional units, and thus appear to lack true segmentation as observed among multiple Cambrian clades. Despite clear axial polarity, there is no evidence for a discrete head with concentrated sensory organs.

A nervous system is an assembly of neurons [71,72]. What constitutes a neuron is less clear [71,73] and such cells are unlikely to be identified in the fossil record. Nervous systems allow rapid communication over significant distances, often integrating multiple sensory inputs to produce a response (e.g. [74]). Sponges lack a nervous system but move and respond to external stimuli. Movement is slow and responses are limited [72,73,75,76]. Placozoans move via epithelia and respond to food availability [77,78] relying on concentrated neurosecretory cells at the body periphery [79].

*Kimberella*, *Ikaria* and *Dickinsonia* fed directly on organic matter covering the Ediacaran seafloor [48,52,55,58] and likely moved to access new resources. Trace fossils associated with *Kimberella* and *Dickinsonia* suggest that they were able to determine when they had consumed sufficient nutrients in a particular area such that movement was more productive than continued feeding. Comparisons with growth rates in modern mats require that, in order to leave traces of such

**Table 1.** Developmental characters of four representative Ediacara taxa and the genetic controls that regulate their formation in modern organisms. 'Basal clade' refers to the earliest appearance of identified regulatory control, in most cases predating the earliest appearance of the character. References cited are for the identification of regulatory control in the basal clade indicated. See also references in [86].

| character | representative White Sea taxa with character | regulatory control | basal clade with regulatory element | references |
|---|---|---|---|---|
| multicellularity | all | actomyosin, cadherins, C-type lectins, LIM Homeobox, Type IV collagens, tyrosine kinases | Holozoa | [22,23,87–89] |
| cell-type differentiation | all | autocrine signalling, microRNAs, Myc, p53, PCR2, SOX/TCF | Holozoa | [23,90] |
| mesoderm | *Kimberella, Ikaria* | β-catenin, Nodal, Notch/Delta | Holozoa | [28,91,92] |
| musculature | *Kimberella, Ikaria, Dickinsonia* | actin, Mef2, MyHC, myocardin | Eukarya | [93,94] |
| axial polarity | all | BMP, Hox, ParaHox, Wnt | Metazoa | [95–99] |
| left–right symmetry | *Kimberella, Ikaria, Dickinsonia* | Nodal | Bilateria | [92] |
| body regionalization (organs, appendages, segmentation, etc.) | *Kimberella*? | CTFCs, distal enhancers, Hox, Notch/Delta, TADs | Metazoa | [90,100–102] |
| nervous system | *Kimberella, Ikaria, Dickinsonia* | bHLH, Notch, SoxB2 | Metazoa | [40,103] |
| CNS | absent | HOX, NK cluster, Nodal, Numb, PAX, Prospero | Bilateria | [40,103,104] |
| immunity | *Dickinsonia* | Toll-like receptors | Holozoa | [24,105] |
| apoptosis | all | Hippo, Myc | Holozoa | [106] |

behaviour, mobility—in some cases extending over several metres of the seafloor—must have occurred relatively recently, likely within hours prior to burial [55]. Restriction of *Helmintho-idichnites* to thin sandstone horizons indicates the chemosensory ability of *Ikaria* to seek out both beneficial oxygenated and toxic sulfidic environments, possibly in response to daily cycles of oxygenic photosynthesis within mats [52,53,80,81].

Ecological similarity between *Dickinsonia* and Placozoa [58] suggests that similar behaviours are possible without a nervous system, although it is unclear if this is scalable to the sizes achieved by *Dickinsonia*. Rapid mobility over large areas and associated with sediment displacement by *Ikaria* is beyond the capacity of Porifera or Placozoa. Burrowing in response to both food availability and environmental suitability suggests a behavioural response integrating distinct sources of information. Systematic excavation observed in scratch marks associated with *Kimberella* [47,48] indicates coordination between a proboscis and 'frill' structure adapted for mobility, separated by centimetres. These features strongly suggest the presence of a nervous system.

These three taxa with possible evidence for neural activity exhibit bilateral symmetry but lack signs of neural condensation. Both morphological and behavioural evidence establishes that complex sensory organs and a CNS were present in the early Cambrian (e.g. [82,83]). Thus, the absence of evidence for a CNS as found in many bilaterians is potentially meaningful.

Future palaeontological studies will likely reveal additional developmentally significant characters. The recent discovery of *Dickinsonia* with morphological defects followed by a return to regular modularity [61,84] indicates repair functions and possibly an immune response. The highly regulated growth and maintenance of constant morphologies in a variety of Ediacara taxa [15,68,85] is difficult to envision without apoptosis.

## (a) Inferred developmental capacity

The developmentally relevant characters described above have conserved regulatory elements that control their expression in modern animals. In table 1 and the discussion below, we use these relationships to identify likely regulatory machinery responsible for their production in representative Ediacara taxa.

Given the assumption of animal affinities, regulatory elements essential for multicellularity and found in holozoans were likely operating in Ediacara taxa. These likely included multiple extracellular matrix domains and TF families, such as cadherins, C-type lectins, tyrosine kinases, LIM Homeobox and canonical Type IV collagens, among others [22,23,86–89].

Different animal cell types are produced via changes to their core regulatory complex of TFs [107]. Other controls, such as microRNAs and autocrine signalling, help maintain individualized cellular identities [24,28,65,108]. Tissue formation builds upon tools involved in the establishment of an extracellular matrix, such as β-catenin [28,91]. These coordinate different life stages in single-celled holozoans (e.g. [23,28,91]). White Sea

taxa likely employed these same genetic elements to produce differentiated cell types and tissue-grade organization.

Actomyosin-based contraction, essential in metazoan musculature, is conserved among eukaryotes for functions including cell division and shape change [93,94]. Thus, common contractile proteins, such as actin and myosin heavy chain, were present in Ediacaran animals. Metazoan lineages constructed individualized TF pathways to build and control muscles, including different muscle types within bilaterian groups [93,94], limiting further classification of muscle-specific gene regulatory pathways.

Axial patterning is normally achieved by antagonistic interactions between morphogenic gradients [95–99,109,110]. A/P differentiation is controlled in bilaterians by the canonical Wnt/β-catenin pathway, with later co-option and expansion by Hox and ParaHox genes [95–99]. Polarized expression of Wnt in non-bilaterian metazoans, including poriferan larvae, ctenophores and cnidarians, suggests a conserved role for these proteins [95,98,111–113]. Antagonistic chordin-BMP signalling for D/V patterning is conserved across bilaterians [95–98,114]. Although homology between cnidarian and bilaterian body axes is unresolved (as discussed by [115]), similar regulatory mechanisms—including Wnt, BMP signalling and Hox genes [111,113,116,117]—likely operated in Ediacaran metazoans with axial differentiation. In modern bilaterians, left–right symmetry, as observed in *Kimberella*, *Ikaria* and *Dickinsonia*, requires the Nodal pathway as an extension of the transforming growth factor (TGF-β) pathway [92].

Increased body regionalization characteristic of bilaterians appears to have required enhanced combinatorial complexity of interactions between existing TFs, and thus deployment of distal enhancers, topologically associated domains (TADs) and insulator proteins (e.g. CTFCs), which jointly structure three-dimensional chromatin interactions [9,28]. For example, Hox genes are integral in the formation of specific anatomical structures such as organs and appendages [118] as well as more basal functions involving axial patterning [95–99,117,119]. Evidence for some degree of gross morphological regionalization in *Kimberella* indicates potential, but limited use of similar regulatory elements.

The Notch receptor and Delta ligand promote cell identity in populations of regionally adjacent cells [90], which Davidson & Erwin described as a reusable 'plug-in' [120]. Although unique to metazoans, these pathways likely evolved from the reshuffling and co-option of protein domains found in single-celled eukaryotes [100]. This signalling pathway is found in non-bilaterian metazoans, for example, in nematocyte and germ cell differentiation in cnidarians [101] and sensory cells in poriferans [121]. Notch/Delta signalling is associated with a range of differentiated systems in bilaterians, including the brain, heart and limbs (see [102] and references therein), apparently absent in the Ediacara Biota. Although patterns of segmentation are variably regulated, a common theme among vertebrates, arthropods and annelids is co-option of Notch/Delta signalling [122]. The absence of a CNS, segmentation and appendages in the Ediacara Biota suggests that Notch/Delta signalling was likely restricted to germ cell differentiation and/or establishment of the nervous system in Ediacara taxa such as *Kimberella* and *Ikaria*. Apparently, many bilaterian co-options had not yet occurred.

Based on common expression in the neuronal regions of ctenophores, cnidarians and bilaterians, the establishment of a nervous system in Ediacara taxa likely involved SoxB2,

Notch and bHLH signalling [40,103,113]. An absence of evidence for the arrangement of this system leaves open the question of whether such taxa used regulatory elements, such as the Nodal pathway, important in establishing neural organization in bilaterians [92].

Immunity was possibly triggered by Toll-like receptors common to cnidarians and bilaterians [105] and recently identified in choanoflagellates [24]. Apoptosis may have been achieved by conserved signalling pathways, such as Hippo, functional in holozoans and used to coordinate cell proliferation and apoptosis in a variety of animals [106].

## (b) Inferred phylogenetic affinities of representative Ediacara taxa

Insights into the developmental capacity of representative Ediacara taxa can be integrated with comparative developmental studies to constrain potential phylogenetic positions (figure 1). Non-metazoans exhibit similar traits to those highlighted here, albeit with different developmental control. Namely, plants are spatially patterned with repeated units oriented relative to the growth axis and controlled via regulatory mechanisms distinct from those used by animals [123,124]. Thus, the identification of specific genetic programming in representative taxa relies on the assumption of metazoan affinity.

Developmental characters are interpreted exclusively from fossil evidence, independent of phylogenetic classification. We consider it most parsimonious that the number of developmentally relevant characters consistent with those found in metazoans add to the growing body of evidence that the Ediacara Biota records the early evolution of animals, rather than the independent evolution of a variety of metazoan features in a 'failed evolutionary experiment' of non-animal taxa [125]. Further, traits identified share many similarities with those of animals not found in other complex organisms. For example, although both use antagonistic gene expression integrating local and global signalling to establish axial polarity, plants maximize morphological flexibility in order to respond to variability in their external environment [124,126]. Animals, as well as the Ediacara Biota, instead use these systems to maintain symmetry and scaling with growth, important for functions such as mobility [124]. Therefore, developmental characters may represent further, independent support for many Ediacara Biota taxa belonging within Metazoa.

Despite unfamiliar body plans, *Tribrachidium* and *Dickinsonia* display cellular and tissue differentiation as well as polarity about at least one body axis. *Tribrachidium* used regulatory programming similar to that present in modern cnidarians. *Dickinsonia* appears to occupy a unique space in metazoan development. It had the capacity to coordinate behaviour across great cellular distances as well as generate polarity, left–right symmetry and patterning relative to the direction of movement, similar to bilaterians. However, there is no evidence for the more complex body regionalization evident in crown bilaterians.

Previous interpretations have indicated that *Ikaria* shares many attributes associated with the PDA (although the PDA probably existed tens of millions of years earlier [1,2]), including the likely presence of a nervous system and a through-gut [52]. *Kimberella* has been allied with Lophotrochozoa [1] and displays axial and regional patterning to produce a proboscis and foot (or analogous structures). These traits are consistent

with the use of bilaterian specific regulatory elements, including β-catenin, distal enhancers, Notch/Delta and Nodal signalling [92].

## 3. Implications for regulatory evolution

There is an absence of evidence for segmentation, appendages, or concentrated sensory organs in representative White Sea taxa. Molecular clocks suggest that numerous bilaterian lineages existed at this time, and thus these characters might be expected [1,2]. Such absence, however, is consistent with phylogenetic and developmental support for morphologically simple basal bilaterians [3,7,9,103,120]. Although speculative, lack of these features may reflect the absence or limited expression of localized boundaries established by systems of cross-repression (for example, Notch/Delta).

Among bilaterians, formation of a discrete head is developmentally decoupled from A/P patterning (e.g. [127]). Axial polarization in Ediacara fossils with no evidence for focused sensory organs is consistent with the independent evolution of a head and ventral nerve cord in several bilaterian clades by co-opting common body patterning submodules [104,127]. An apparent unifying feature of bilaterian cephalization, and similar structures such as lophophores, is a lack of Hox expression and deactivation of Wnt signalling via antagonistic gene expression [127,128]. The hypothesized involvement of Hox and Wnt in the development of *Dickinsonia*, *Ikaria* and *Kimberella* is consistent with the use of these genetic controls prior to the anterior concentration of sensory organs.

Organization of the CNS in bilaterians is related to A/P and D/V patterning, with developmental control involving Hox genes, Wnt and BMP pathways [40,103,104,113,129]. However, cnidarians and some bilaterians (e.g. Xenacoelomorpha) exhibit axial patterning and a non-centralized nervous system [103,104,117]. We propose that several Ediacara taxa represent lineages that similarly had not co-opted these pathways for more complex regionalization. While it is possible that a CNS was present in the PDA and that this absence represents subsequent loss, we consider this unlikely as it would require the earliest bilaterians in the fossil record to be highly derived relative to this common ancestor. Our proposal is consistent with the hypothesis that rudimentary nervous systems were present in early metazoan lineages and persisted for a long period before multiple independent origins of a CNS [7].

## 4. Concluding remarks

We have evaluated the developmental capacity of representative White Sea taxa, identified several metazoan-specific morphogenetic processes and the likely regulatory elements responsible for their expression. This allows us to bound potential phylogenetic positions of these taxa relative to extant metazoans. At least three occupy the significant gap between the ability to produce body polarization and a nervous system and the subsequent evolutionary adaptations required for more complex regionalization and the formation of a CNS. Although diverse bilaterian body plans do not appear until the Cambrian, bilaterians and gene regulatory elements critical for their later success were represented in the Ediacara Biota. Future work focused on resolving additional developmentally important characters in Ediacara taxa (e.g. morphogenesis [15]), incorporating novel discoveries of gene regulatory networks in modern organisms and identifying variations through time will further refine our understanding of early animal evolution and diversification.

Data accessibility. This article has no additional data.

Authors' contributions. S.D.E., M.L.D. and D.H.E. conceived of this study and edited the manuscript, S.D.E. and D.H.E. composed the manuscript and S.D.E. created the figures and table.

Competing interests. We declare we have no competing interests.

Funding. This work was supported by a Peter Buck postdoctoral fellowship to S.D.E. and a NASA Exobiology Grant no. (NNX14AJ86G) to M.L.D.

Acknowledgements. We thank Jim Gehling, Christine Hall, Ian Hughes and Karma Nanglu for insightful discussion while preparing this manuscript, Michelle Droser for providing line drawings of representative taxa used in figure 1, and the reviewers for useful comments.

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
