## [Peer Review File · Proceedings of the Royal Society B: Biological Sciences]

Review History

RSPB-2020-2334.R0 (Original submission)

Review form: Reviewer 1 (Arsham Nejad Kourki)

Recommendation

Major revision is needed (please make suggestions in comments)

Scientific importance: Is the manuscript an original and important contribution to its field?

Acceptable

General interest: Is the paper of sufficient general interest?

Good

Quality of the paper: Is the overall quality of the paper suitable?

Marginal

Is the length of the paper justified?

Yes

Should the paper be seen by a specialist statistical reviewer?

No

Do you have any concerns about statistical analyses in this paper? If so, please specify them explicitly in your report.

No

It is a condition of publication that authors make their supporting data, code and materials available - either as supplementary material or hosted in an external repository. Please rate, if applicable, the supporting data on the following criteria.

Is it accessible?

N/A

Is it clear?

N/A

Is it adequate?

N/A

Do you have any ethical concerns with this paper?

No

Comments to the Author

This paper touches on a very interesting subject which has been stimulated by recent empirical findings. It is a bold attempt at integrating evidence from developmental biology and palaeontology in order to shed light on metazoan phylogeny and character evolution. However, it suffers from a number of serious issues:

1. A considerable number of the inferences made in the paper aren't sufficiently supported by the evidence presented/aren't sufficiently well-argued.
2. The overall argument of the paper is, partly as a result of the weakness of smaller inferences, not very convincing. Furthermore, a big part of the argument is redundant.
3. The paper doesn't exactly achieve what it aims at in the introduction, and overblows what it has achieved in the conclusion.
4. The figures are of very low quality.
5. There are numerous mistakes in punctuation, and at least one in grammar.

Please see attached pdf for detailed comments. I have not pointed out issue 5 in the comments.

Review form: Reviewer 2

Recommendation

Major revision is needed (please make suggestions in comments)

Scientific importance: Is the manuscript an original and important contribution to its field?

Acceptable

General interest: Is the paper of sufficient general interest?

Acceptable

Quality of the paper: Is the overall quality of the paper suitable?

Acceptable

Is the length of the paper justified?

Yes

Should the paper be seen by a specialist statistical reviewer?

No

Do you have any concerns about statistical analyses in this paper? If so, please specify them explicitly in your report.

No

It is a condition of publication that authors make their supporting data, code and materials available - either as supplementary material or hosted in an external repository. Please rate, if applicable, the supporting data on the following criteria.

Is it accessible?

N/A

Is it clear?

N/A

Is it adequate?

N/A

Do you have any ethical concerns with this paper?

No

Comments to the Author

The article by Evans et al "Developmental processes in Ediacaran macrofossils" is an interesting idea to use developmental characters to place ediacarians in the Metazoan phylogeny and also to characterize the fossilized. It is NOT a research paper, as it has been categorized, it is a review or perspective. So it should be declared as this, since there is no new data provided.

I find this approach also a bit circulatory, reminding me on the old days where animal phylogenies were purely based on morphological characters. However, since we deal with fossils here and the animal phylogeny provides a relatively solid frame, it is ok to proceed like this. The whole paper lacks at some parts the necessary accuracy in the description of the molecular components. The authors need to rework the wording and make it more specific and expand the categories, such as structural proteins (which is NOT a pathway), transcription factors and signalling cascades in Table 1. Furthermore, I find the references a bit randomly chosen and some of the key papers are not listed (e.g. the Hox in *Nematostella* DOI: 10.1126/science.aar8384, the actin-myosin presence and function in choanoflagellates has been shown much earlier than Brunet et al). The manuscript would gain by adding more references and reworking the present. I am also not friend with the Figures. Figure 1 is ok and necessary, but I don't know the function of Figure 2 and I do not think it is necessary at all. Figure 3 has a very ladder-like impression, picturing the great chain of beings and must be reworked. It also clearly can be made more attractive.

See below some remarks to the characters:

Multicellularity

Coming with the assumption that the ediacarians are Metazoa, it is obvious to assume their multicellularity. The authors should maybe include the hallmark and apomorphy of Metazoa, which is the sperm and oocyte production through meiosis. Also means the germ line, likely specified through PIWI etc.

Anterior-Posterior and Dorsal-Ventral polarity

I suggest the authors change it to perpendicular body axes. It can still be related to the same molecules, but even in Cnidaria it is not clear what axis can be related to the Bilateria. E.g. the BMP-chordin axis is the same as the Hox-axis and even connected through regulation (see <http://dx.doi.org/10.1016/j.celrep.2015.02.035>). The Wnt axis in cnidarians is perpendicular to this. Therefore it is to assume that the axial patterning systems were present, but they can not be

related to A-P or D-V. So the authors should change this and it also makes the whole section of guessing what is D-V or A-P in these animals obsolete.

Differentiation

- The authors state that “coeloms” are present possibly in Kimberella and Ikaria (line 208) and connect this to the establishment of three germ layers. However, it is not necessarily the case. there are many kinds of cavities in animals that can be formed without the presence of three germ layers (gastric pouches in cnidaria, other cavities in ctenophores, pseudocoeloms etc.). So it is not necessarily the case. Here, in this work, the authors aim and should aim for the minimal presence of characters and try to avoid overestimations.

The role of brachyury is not clear. What is clear is, that it has many functions in cell internalization (not cell migration), not necessarily connected to mesoderm. I suggest to eliminate brachyury from the whole paper, because its pure speculation.

Line 240: Hox are not “establishing” polarity, but pattern along the axis. Polarity is established much earlier, namely by Wnt in most cases.

Nervous System:

In general I feel that Placozoans are not considered sufficiently in the manuscript. It is to assume that placozoans can sense their environment without nervous system. I think they play an important role in the section of the nervous system as do carnivorous sponges for example.

Review form: Reviewer 3

Recommendation

Reject – article is scientifically unsound

Scientific importance: Is the manuscript an original and important contribution to its field?

Marginal

General interest: Is the paper of sufficient general interest?

Good

Quality of the paper: Is the overall quality of the paper suitable?

Poor

Is the length of the paper justified?

Yes

Should the paper be seen by a specialist statistical reviewer?

No

Do you have any concerns about statistical analyses in this paper? If so, please specify them explicitly in your report.

No

It is a condition of publication that authors make their supporting data, code and materials available - either as supplementary material or hosted in an external repository. Please rate, if applicable, the supporting data on the following criteria.

Is it accessible?

N/A

Is it clear?

N/A

Is it adequate?

N/A

Do you have any ethical concerns with this paper?

No

Comments to the Author

Evans and colleagues characterise key features of a select suite of fossil organisms from the Ediacaran macrobiota, aiming to identify key anatomical features that may be linked to molecular genetic developmental processes of animal bodyplan organisation - which they review. Their explicit aims are to:

"First, by integrating insights from the fossil record with comparative data from living taxa, can reliable inferences be made about the developmental capacity of Ediacaran clades?"

"Second, can we infer plausible phylogenetic positions for and/or relationships between these taxa with the addition of this developmental information?"

They conclude that the considered elements of the White Sea macrobiota provide evidence for "several metazoan specific regulatory elements, suggesting that the Ediacara Biota included animals (e.g. [1, 11]). Our analysis further supports that many of these organisms occupy the significant gap between the adaptation of signalling pathways responsible for body polarization and nervous system development and their subsequent co-option for more specific regionalization and the formation of a CNS."

I am extremely sympathetic to the aims of this study, however, I am concerned that, as presented, it lacks a logical basis for inference. Specifically, the search for evidence of metazoan developmental mechanisms assumes that these organisms are animals and, yet, this is presented as a conclusion. The fossils are used to infer the sequential acquisition of developmental mechanisms as well as the phylogenetic affinity of the organisms based on the anatomical characteristics, rooted in developmental mechanisms, that the fossils exhibit; it is not possible to do the former without the latter being established a priori.

In summary, the logic presented in the manuscript reads as circular. Much of the text is review rather results and it is not clear that the review yields any added value. I would suggest that the manuscript/study is revised with a more logical inference framework:

1. Set out the phylogenetic affinity of living clades first
2. Infer anatomical character evolution within this framework
3. Review regulatory evolution within this framework
4. Resolve affinity of fossils in this framework
5. Evaluate what impact fossils have on inferences of anatomical character and regulatory evolution

This structure includes many of the same components of the current manuscript but, crucially, the process of inference is linear; assumptions and insights are more readily distinguishable.

At present, the absence of an explicit phylogenetic hypothesis of living organisms is a major limitation. How are the described clades defined? Does Bilateria equate to Nephrozoa or does it also encompass Xen/aceolomorpha? Where do the authors resolve ctenophores and, if they consider the position of ctenophores uncertain, what impact does this have on their analysis?

Crucially, what are the characters that supports the assignment of the fossils to specific positions within the phylogenetic tree?

Details

Lines 51-53: Many taxa from the Ediacara Biota may represent stem lineages of major animal groups but their diagnostic characters either were not preserved or had not yet evolved.

Be specific WRT to 'stem lineages of major animal groups' since Metazoa and Eumetazoa and Bilateria are major animal groups, the diagnostic characters of which are preserved, hence this study. I think, rather, that the authors are referring to phyla and their constituent clades.

Lines 147-152: Given similarities with animals, protein domains essential for multicellularity found in holozoans and basal metazoan clades were likely present in Ediacaran taxa. For example, actomyosin-based contractility of cell sheets (necessary for epithelia formation) has been described in choanoflagellates [35] and ichthysporeans [70]. Regulatory elements likely included multiple extracellular matrix domains and transcription factor families, such as cadherins, C-type lectins, tyrosine kinases, LIM Homeobox and canonical Type IV collagens, among others [36-38].

Do you mean protein domains (the structural modular components of proteins) or protein families? Why are these likely present in Ediacaran taxa? Unless you assume that these fossil organisms are crown holozoans? In which case, what is the insight?

Lines 157-159: Although polarization in Dickinsonia may not be homologous to the A/P axis in bilaterians [11], it possessed the developmental capacity to produce an A/P axis and differentiate between two "ends" relative to the direction of movement [26].

How does this follow? Why not another axis?

Lines 180-182: We consider the surface preserved facing upwards into the overlying sand as the top or dorsal surface and the opposite as bottom or ventral. As with A/P polarity, it is the ability to differentiate these that is important here.

What are we learning here? Affinity or phylogenetic position of the fossil? When the mechanism evolved?

Lines 199-214: Differentiation

The scale and patterning of numerous Ediacara Biota taxa implies multiple cell types and some degree of regional differentiation. Early forms, such as fractally organized rangeomorphs, may have been possible with a few cell types, although a recent reconstruction suggests at least a differentiated epithelium (Butterfield, 2020). Differentiation of White Sea taxa, such as Dickinsonia, is less clear, although an undivided anterior region (Fig 2a) and possible musculature [24, 26, 27] (see below) support some level of cellular disparity. Kimberella has perhaps the strongest evidence for distinct cell-types and regional specification of functional units (Fig 2b). The ability to excavate the organic mat and displace sediment grains in Kimberella and Ikaria supports the presence of a coelom [21, 74]. This is unique to bilaterians and signifies the establishment of three distinct germ layers. Further, feeding in Kimberella [17, 18] and Ikaria [21, 22] suggests that they had a mouth and some type of gut, potentially a through-gut, although direct morphological evidence for these structures has yet to be recognized. While coeloms evolved several times among bilaterians, the presence of coelom and gut required the ability to form multiple tissue layers.

Cell types or tissues or organs? Epithelia only require cell polarization, not necessarily differentiation of cell types. Organs require epithelia. Focus on what you are inferring.

Musculature: locomotion. Why stray into other taxa like Haootia? Are you considering them together because you think they are a clade and provide reciprocal insights?

CNS complexity: evidence of absence or absence of evidence?

Problematic characters: sexual reproduction rests too much on inference from size classes

Discussion:

Inferences of characters have already assumed affinity, to some extent at least, and so you cannot use these inferences to infer phylogenetic position, either of the fossils or the timing of origin of regulatory factors.

This discussion does not build on what goes before; does not formally interpret the 'results'.

Decision letter (RSPB-2020-2334.R0)

12-Oct-2020

Dear Dr Evans:

I am writing to inform you that your manuscript RSPB-2020-2334 entitled "Developmental processes in Ediacaran macrofossils" has, in its current form, been rejected for publication in Proceedings B.

This action has been taken on the advice of referees, who have recommended that substantial revisions are necessary. With this in mind we would be happy to consider a resubmission, provided the comments of the referees are fully addressed. However please note that this is not a provisional acceptance.

Please note that this decision may (or may not) have taken into account confidential comments.

In your revision process, please take a second look at how open your science is; our policy is that *ALL* (maximally inclusive) data involved with the study should be made openly accessible,

fully enabling re-use, replication and transparency-- see:
<https://royalsociety.org/journals/ethics-policies/data-sharing-mining/>
 Insufficient sharing of data can delay or even cause rejection of a paper.
 Full data and code/scripts to enable reuse/replication/repurposing are what this policy intends.

Sincerely,
 Dr John Hutchinson, Editor
 mailto: proceedingsb@royalsociety.org

Associate Editor
 Board Member: 1

Comments to Author:

The paper has been viewed by three reviewers. While all have raised some points of remarks as to the interesting idea in this endeavour, there are issues raised in terms of the logic of the study and its nature of being circular in its sequence of inference.

Based on this I cannot recommend the study to be published in its current form. It will need a dramatic revision to acknowledge the logic of sequence and mode of inference. Alternatively it will stick to a more review-style nature and instead pose the likelihood that developmentally important transcription factors can be surmised to have been established among different lineages of Ediacaran macrofossils based on the developmental mode and likely place in the eumetazoan total group. However, it will need to have a more cautionary slant to it.

The literature review is inadequate in terms of presenting the most relevant papers for particular findings and statements.

On a more personal note, I think that the illustrations in the main paper is highly inadequate in terms of image quality and the highly schematic nature of the diagrams of body regions in relation to the nature of the questions raised about body axis patterning by transcription factors and relating the nature of these body plans in the Ediacaran to modern relatives.

Reviewer(s)' Comments to Author:

Referee: 1

Comments to the Author(s)

This paper touches on a very interesting subject which has been stimulated by recent empirical findings. It is a bold attempt at integrating evidence from developmental biology and palaeontology in order to shed light on metazoan phylogeny and character evolution. However, it suffers from a number of serious issues:

1. A considerable number of the inferences made in the paper aren't sufficiently supported by the evidence presented/aren't sufficiently well-argued.
2. The overall argument of the paper is, partly as a result of the weakness of smaller inferences, not very convincing. Furthermore, a big part of the argument is redundant.
3. The paper doesn't exactly achieve what it aims at in the introduction, and overblows what it has achieved in the conclusion.
4. The figures are of very low quality.
5. There are numerous mistakes in punctuation, and at least one in grammar.

Please see attached pdf for detailed comments. I have not pointed out issue 5 in the comments.

Referee: 2

Comments to the Author(s)

The article by Evans et al "Developmental processes in Ediacaran macrofossils" is an interesting idea to use developmental characters to place ediacarians in the Metazoan phylogeny and also to

characterize the fossilized. It is NOT a research paper, as it has been categorized, it is a review or perspective. So it should be declared as this, since there is no new data provided.

I find this approach also a bit circulatory, reminding me on the old days where animal phylogenies were purely based on morphological characters. However, since we deal with fossils here and the animal phylogeny provides a relatively solid frame, it is ok to proceed like this. The whole paper lacks at some parts the necessary accuracy in the description of the molecular components. The authors need to rework the wording and make it more specific and expand the categories, such as structural proteins (which is NOT a pathway), transcription factors and signalling cascades in Table 1. Furthermore, I find the references a bit randomly chosen and some of the key papers are not listed (e.g. the Hox in *Nematostella* DOI: 10.1126/science.aar8384, the actin-myosin presence and function in choanoflagellates has been shown much earlier than Brunet et al). The manuscript would gain by adding more references and reworking the present. I am also not friend with the Figures. Figure 1 is ok and necessary, but I don't know the function of Figure 2 and I do not think it is necessary at all. Figure 3 has a very ladder-like impression, picturing the great chain of beings and must be reworked. It also clearly can be made more attractive.

See below some remarks to the characters:

Multicellularity

Coming with the assumption that the ediacarians are Metazoa, it is obvious to assume their multicellularity. The authors should maybe include the hallmark and apomorphy of Metazoa, which is the sperm and oocyte production through meiosis. Also means the germ line, likely specified through PIWI etc.

Anterior-Posterior and Dorsal-Ventral polarity

I suggest the authors change it to perpendicular body axes. It can still be related to the same molecules, but even in Cnidaria it is not clear what axis can be related to the Bilateria. E.g. the BMP-chordin axis is the same as the Hox-axis and even connected through regulation (see <http://dx.doi.org/10.1016/j.celrep.2015.02.035>). The Wnt axis in cnidarians is perpendicular to this. Therefore it is to assume that the axial patterning systems were present, but they can not be related to A-P or D-V. So the authors should change this and it also makes the whole section of guessing what is D-V or A-P in these animals obsolete.

Differentiation

- The authors state that "coeloms" are present possibly in *Kimberella* and *Ikaria* (line 208) and connect this to the establishment of three germ layers. However, it is not necessarily the case. There are many kinds of cavities in animals that can be formed without the presence of three germ layers (gastric pouches in cnidaria, other cavities in ctenophores, pseudocoeloms etc.). So it is not necessarily the case. Here, in this work, the authors aim and should aim for the minimal presence of characters and try to avoid overestimations.

The role of brachyury is not clear. What is clear is, that it has many functions in cell internalization (not cell migration), not necessarily connected to mesoderm. I suggest to eliminate brachyury from the whole paper, because its pure speculation.

Line 240: Hox are not "establishing" polarity, but pattern along the axis. Polarity is established much earlier, namely by Wnt in most cases.

Nervous System:

In general I feel that Placozoans are not considered sufficiently in the manuscript. It is to assume that placozoans can sense their environment without nervous system. I think they play an important role in the section of the nervous system as do carnivorous sponges for example.

Referee: 3

Comments to the Author(s)

Evans and colleagues characterise key features of a select suite of fossil organisms from the Ediacaran macrobiota, aiming to identify key anatomical features that may be linked to molecular genetic developmental processes of animal bodyplan organisation - which they review. Their explicit aims are to:

"First, by integrating insights from the fossil record with comparative data from living taxa, can reliable inferences be made about the developmental capacity of Ediacaran clades?"

"Second, can we infer plausible phylogenetic positions for and/or relationships between these taxa with the addition of this developmental information?"

They conclude that the considered elements of the White Sea macrobiota provide evidence for "several metazoan specific regulatory elements, suggesting that the Ediacara Biota included animals (e.g. [1, 11]). Our analysis further supports that many of these organisms occupy the significant gap between the adaptation of signalling pathways responsible for body polarization and nervous system development and their subsequent co-option for more specific regionalization and the formation of a CNS."

I am extremely sympathetic to the aims of this study, however, I am concerned that, as presented, it lacks a logical basis for inference. Specifically, the search for evidence of metazoan developmental mechanisms assumes that these organisms are animals and, yet, this is presented as a conclusion. The fossils are used to infer the sequential acquisition of developmental mechanisms as well as the phylogenetic affinity of the organisms based on the anatomical characteristics, rooted in developmental mechanisms, that the fossils exhibit; it is not possible to do the former without the latter being established a priori.

In summary, the logic presented in the manuscript reads as circular. Much of the text is review rather results and it is not clear that the review yields any added value. I would suggest that the manuscript/study is revised with a more logical inference framework:

1. Set out the phylogenetic affinity of living clades first
2. Infer anatomical character evolution within this framework
3. Review regulatory evolution within this framework
4. Resolve affinity of fossils in this framework
5. Evaluate what impact fossils have on inferences of anatomical character and regulatory evolution

This structure includes many of the same components of the current manuscript but, crucially, the process of inference is linear; assumptions and insights are more readily distinguishable.

At present, the absence of an explicit phylogenetic hypothesis of living organisms is a major limitation. How are the described clades defined? Does Bilateria equate to Nephrozoa or does it also encompass Xen/aceolomorpha? Where do the authors resolve ctenophores and, if they consider the position of ctenophores uncertain, what impact does this have on their analysis?

Crucially, what are the characters that supports the assignment of the fossils to specific positions within the phylogenetic tree?

Details

Lines 51-53: Many taxa from the Ediacara Biota may represent stem lineages of major animal groups but their diagnostic characters either were not preserved or had not yet evolved.

Be specific WRT to 'stem lineages of major animal groups' since Metazoa and Eumetazoa and Bilateria are major animal groups, the diagnostic characters of which are preserved, hence this study. I think, rather, that the authors are referring to phyla and their constituent clades.

Lines 147-152: Given similarities with animals, protein domains essential for multicellularity found in holozoans and basal metazoan clades were likely present in Ediacaran taxa. For example, actomyosin-based contractility of cell sheets (necessary for epithelia formation) has been described in choanoflagellates [35] and ichthysporeans [70]. Regulatory elements likely included multiple extracellular matrix domains and transcription factor families, such as cadherins, C-type lectins, tyrosine kinases, LIM Homeobox and canonical Type IV collagens, among others [36-38].

Do you mean protein domains (the structural modular components of proteins) or protein families? Why are these likely present in Ediacaran taxa? Unless you assume that these fossil organisms are crown holozoans? In which case, what is the insight?

Lines 157-159: Although polarization in Dickinsonia may not be homologous to the A/P axis in bilaterians [11], it possessed the developmental capacity to produce an A/P axis and differentiate between two "ends" relative to the direction of movement [26].

How does this follow? Why not another axis?

Lines 180-182: We consider the surface preserved facing upwards into the overlying sand as the top or dorsal surface and the opposite as bottom or ventral. As with A/P polarity, it is the ability to differentiate these that is important here.

What are we learning here? Affinity or phylogenetic position of the fossil? When the mechanism evolved?

Lines 199-214: Differentiation

The scale and patterning of numerous Ediacara Biota taxa implies multiple cell types and some degree of regional differentiation. Early forms, such as fractally organized rangeomorphs, may have been possible with a few cell types, although a recent reconstruction suggests at least a differentiated epithelium (Butterfield, 2020). Differentiation of White Sea taxa, such as Dickinsonia, is less clear, although an undivided anterior region (Fig 2a) and possible musculature [24, 26, 27] (see below) support some level of cellular disparity. Kimberella has perhaps the strongest evidence for distinct cell-types and regional specification of functional units (Fig 2b). The ability to excavate the organic mat and displace sediment grains in Kimberella and Ikaria supports the presence of a coelom [21, 74]. This is unique to bilaterians and signifies the establishment of three distinct germ layers. Further, feeding in Kimberella [17, 18] and Ikaria [21, 22] suggests that they had a mouth and some type of gut, potentially a through-gut, although direct morphological evidence for these structures has yet to be recognized. While coeloms evolved several times among bilaterians, the presence of coelom and gut required the ability to form multiple tissue layers.

Cell types or tissues or organs? Epithelia only require cell polarization, not necessarily differentiation of cell types. Organs require epithelia. Focus on what you are inferring.

Musculature: locomotion. Why stray into other taxa like Haootia? Are you considering them together because you think they are a clade and provide reciprocal insights?

CNS complexity: evidence of absence or absence of evidence?

Problematic characters: sexual reproduction rests too much on inference from size classes

Discussion:

Inferences of characters have already assumed affinity, to some extent at least, and so you cannot use these inferences infer phylogenetic position, either of the fossils or the timing of origin of regulatory factors.

This discussion does not build on what goes before; does not formally interpret the 'results'.

Author's Response to Decision Letter for (RSPB-2020-2334.R0)

See Appendix A.

RSPB-2020-3055.R0

Review form: Reviewer 1

Recommendation

Accept as is

Scientific importance: Is the manuscript an original and important contribution to its field?

Good

General interest: Is the paper of sufficient general interest?

Good

Quality of the paper: Is the overall quality of the paper suitable?

Good

Is the length of the paper justified?

Yes

Should the paper be seen by a specialist statistical reviewer?

No

Do you have any concerns about statistical analyses in this paper? If so, please specify them explicitly in your report.

No

It is a condition of publication that authors make their supporting data, code and materials available - either as supplementary material or hosted in an external repository. Please rate, if applicable, the supporting data on the following criteria.

Is it accessible?

N/A

Is it clear?

N/A

Is it adequate?

N/A

Do you have any ethical concerns with this paper?

No

Comments to the Author

This manuscript is significantly better-developed than the previous one, and I find it sufficiently well-developed for publication in this journal. I would like to thank the authors for incorporating the (hopefully constructive) comments.

Review form: Reviewer 3

Recommendation

Accept as is

Scientific importance: Is the manuscript an original and important contribution to its field?

Acceptable

General interest: Is the paper of sufficient general interest?

Excellent

Quality of the paper: Is the overall quality of the paper suitable?

Acceptable

Is the length of the paper justified?

Yes

Should the paper be seen by a specialist statistical reviewer?

No

Do you have any concerns about statistical analyses in this paper? If so, please specify them explicitly in your report.

No

It is a condition of publication that authors make their supporting data, code and materials available - either as supplementary material or hosted in an external repository. Please rate, if applicable, the supporting data on the following criteria.

Is it accessible?

N/A

Is it clear?

N/A

Is it adequate?

N/A

Do you have any ethical concerns with this paper?

No

Comments to the Author

The manuscript is much improved. I remain a little concerned that the underpinning logic is unclear - do the fossils constrain the evolution of molecular and developmental traits or does the inference of molecular and developmental traits in the fossils constrain their affinity - can it be both? The initial survey of phenotypic traits does most of the work in itself and hence figure 1. Maybe it would be better to integrate the inference of phenotypic, molecular genetic and

developmental traits into one section?

Regardless, this manuscript works much better as a review: it is provocative, interesting and interdisciplinary.

Decision letter (RSPB-2020-3055.R0)

25-Jan-2021

Dear Dr Evans

I am pleased to inform you that your Review manuscript RSPB-2020-3055 entitled "Developmental processes in Ediacaran macrofossils" has been accepted for publication in Proceedings B. Congratulations!!

Reviewers were won over by the revised MS, even if they do not find it 100% convincing; they agree it is publishable.

The referee(s) do not recommend any further changes. Therefore, please proof-read your manuscript carefully and upload your final files for publication. Because the schedule for publication is very tight, it is a condition of publication that you submit the revised version of your manuscript within 7 days. If you do not think you will be able to meet this date please let me know immediately.

To upload your manuscript, log into <http://mc.manuscriptcentral.com/prsb> and enter your Author Centre, where you will find your manuscript title listed under "Manuscripts with Decisions." Under "Actions," click on "Create a Revision." Your manuscript number has been appended to denote a revision.

You will be unable to make your revisions on the originally submitted version of the manuscript. Instead, upload a new version through your Author Centre.

- 1) A text file of the manuscript (doc, txt, rtf or tex), including the references, tables (including captions) and figure captions. Please remove any tracked changes from the text before submission. PDF files are not an accepted format for the "Main Document".
- 2) A separate electronic file of each figure (tiff, EPS or print-quality PDF preferred). The format should be produced directly from original creation package, or original software format. Please note that PowerPoint files are not accepted.
- 3) Electronic supplementary material: this should be contained in a separate file from the main text and the file name should contain the author's name and journal name, e.g. `authorname_procb_ESM_figures.pdf`

All supplementary materials accompanying an accepted article will be treated as in their final form. They will be published alongside the paper on the journal website and posted on the online figshare repository. Files on figshare will be made available approximately one week before the accompanying article so that the supplementary material can be attributed a unique DOI. Please see: <https://royalsociety.org/journals/authors/author-guidelines/>

- 4) Data-Sharing and data citation

It is a condition of publication that data supporting your paper are made available. Data should be made available either in the electronic supplementary material or through an appropriate repository. Details of how to access data should be included in your paper. Please see <https://royalsociety.org/journals/ethics-policies/data-sharing-mining/> for more details.

If you wish to submit your data to Dryad (<http://datadryad.org/>) and have not already done so you can submit your data via this link <http://datadryad.org/submit?journalID=RSPB&manu=RSPB-2020-3055> which will take you to your unique entry in the Dryad repository.

Once again, thank you for submitting your manuscript to Proceedings B and I look forward to receiving your final version. If you have any questions at all, please do not hesitate to get in touch.

Sincerely,

Associate Editor

Reviewer(s)' Comments to Author:

Referee: 3

Comments to the Author(s)

The manuscript is much improved. I remain a little concerned that the underpinning logic is unclear - do the fossils constrain the evolution of molecular and developmental traits or does the inference of molecular and developmental traits in the fossils constrain their affinity - can it be both? The initial survey of phenotypic traits does most of the work in itself and hence figure 1. Maybe it would be better to integrate the inference of phenotypic, molecular genetic and developmental traits into one section?

Regardless, this manuscript works much better as a review: it is provocative, interesting and interdisciplinary.

Referee: 1

Comments to the Author(s)

This manuscript is significantly better-developed than the previous one, and I find it sufficiently well-developed for publication in this journal. I would like to thank the authors for incorporating the (hopefully constructive) comments.

Decision letter (RSPB-2020-3055.R1)

01-Feb-2021

Dear Dr Evans

I am pleased to inform you that your manuscript entitled "Developmental processes in Ediacaran macrofossils" has been accepted for publication in Proceedings B.

If you are likely to be away from e-mail contact during this period, let us know. Due to rapid publication and an extremely tight schedule, if comments are not received, we may publish the paper as it stands.

Open access

You are invited to opt for open access via our author pays publishing model. Payment of open access fees will enable your article to be made freely available via the Royal Society website as soon as it is ready for publication. For more information about open access publishing please visit our website at http://royalsocietypublishing.org/site/authors/open_access.xhtml.

The open access fee is £1,700 per article (plus VAT for authors within the EU). If you wish to opt for open access then please let us know as soon as possible.

Paper charges

Sincerely,
Proceedings B
<mailto:proceedingsb@royalsociety.org>

Smithsonian

National Museum of Natural History

Department of Paleobiology

Appendix A

7 December 2020

Dear *Proceedings B* editors,

Please find enclosed our revised manuscript: “Developmental processes in Ediacaran macrofossils” by Scott D. Evans (corresponding author), Mary L. Droser and Douglas H. Erwin for consideration of publication in *Proceedings of the Royal Society B, Biological Sciences*. Our resubmission includes the revised manuscript text (both clean and tracked changes versions), two in text figures and one table. This represents the resubmission of a previous manuscript (RSPB-2020-2334). Below you will find our responses to comments by the board member and three anonymous reviewers.

We are grateful to all who offered comment on this work, and the manuscript has been thoroughly reorganized and the goals clarified in response to these comments. We are especially appreciative of the organization proposed by reviewer #3, which we have adopted and feel greatly improves the arguments from our original submission. Suggested changes to the figures improve both the information they provide as well as previous stylistic shortcomings.

Despite major revisions, we note that the abstract of this manuscript has remained largely unchanged. We feel this reflects the strength of this contribution, highlighted by the board member and all reviewers. Namely, that the identification of developmentally relevant characters in representative Ediacara taxa provides a refined understanding of the genetic control responsible for and the phylogenetic placement of this biota. We recognize that this relies on earlier descriptions of representative taxa and that, as such, may be better classified as a review. However, we are unaware of any previous attempts to describe a comprehensive suite of underlying genetic mechanisms responsible for features of the Ediacara Biota as attempted in this work. Further, this approach reveals both the likely presence (e.g. nervous system) and absence (e.g. concentrated sensory machinery) of key characters in a variety of early eumetazoans. This will be of broad scientific interest with implications ranging from the apparent disconnect between the Ediacaran and Cambrian fossil records, to the developmental underpinnings of animal evolution.

Please do not hesitate to contact me with any questions.

Respectfully submitted,

Scott Evans (on behalf of all of the authors)

Response to Reviewer Comments

Please find below our responses (**blue text**) to reviewer comments (black text). Excerpts from the text are presented in “*blue italics*”, although given the extensive nature of reviewer comments and our responses we have largely omitted such excerpts and reference specific passages from the text by line number, referring to the clean version of the manuscript.

Board Member: 1

Comments to Author:

The paper has been viewed by three reviewers. While all have raised some points of remarks as to the interesting idea in this endeavour, there are issues raised in terms of the logic of the study and its nature of being circular in its sequence of inference.

Based on this I cannot recommend the study to be published in its current form. It will need a dramatic revision to acknowledge the logic of sequence and mode of inference. Alternatively it will stick to a more review-style nature and instead pose the likelihood that developmentally important transcription factors can be surmised to have been established among different lineages of Ediacaran macrofossils based on the developmental mode and likely place in the eumetazoan total group. However, it will need to have a more cautionary slant to it.

Based on comments by all three reviewers and highlighted by the board member here, we have significantly revised this manuscript to follow the structure proposed by reviewer 3. We believe these comments have allowed us to significantly strengthen the arguments in the manuscript. Importantly, the description of this contribution above, and emphasized by reviewers below, is exactly the aim of this work and we feel that edits now present this in a logical and more cautious manner without compromising the significance of our work. We are happy for this manuscript to be classified as a review.

The literature review is inadequate in terms of presenting the most relevant papers for particular findings and statements.

We have added 38 new references to the manuscript both in response to specific reviewer comments and to address this concern more generally (as a result of proposed revisions 23 references from the original manuscript are no longer included). While we could have added a number of other references, we are also cognizant of length limitations. We are happy to add any further references the reviewers or editors feel have been omitted.

On a more personal note, I think that the illustrations in the main paper is highly inadequate in terms of image quality and the highly schematic nature of the diagrams of body regions in relation to the nature of the questions raised about body axis patterning by transcription factors and relating the nature of these body plans in the Ediacaran to modern relatives.

In response to this comment and comments by all three reviewers we have removed our original Fig 2 and significantly modified Figs 1 (now Fig 2) and 3 (now Fig 1). Specifically, our figure highlighting morphological characters of representative Ediacara taxa now includes a more

illustrative photograph of *Tribrachidium*, labels of relevant features as well as trace fossils associated with *Kimberella* and *Ikaria* and for the direction of movement in *Dickinsonia* so that it is more accessible to those unfamiliar with the fossil record of these organisms (per reviewer 1 comments). We have revised our phylogeny to more inclusively present metazoan relationships and improved line drawings of representative Ediacara taxa, which we feel more appropriately highlights the affinities of these taxa based on developmental characters and underlying regulatory control.

Reviewer 1

This paper touches on a very interesting subject which has been stimulated by recent empirical findings. It is a bold attempt at integrating evidence from developmental biology and palaeontology in order to shed light on metazoan phylogeny and character evolution. However, it suffers from a number of serious issues:

1. A considerable number of the inferences made in the paper aren't sufficiently supported by the evidence presented/aren't sufficiently well-argued.

We have made significant revisions to address this comment. In particular we have expanded discussion of the evidence for cellular and tissue level differentiation, axial polarity and a nervous system.

2. The overall argument of the paper is, partly as a result of the weakness of smaller inferences, not very convincing. Furthermore, a big part of the argument is redundant.

As suggested by all reviewers and the board member, we feel that the new organization of the manuscript, beginning with the assumption of metazoan affinities, presents a more convincing argument and reduces redundancies.

3. The paper doesn't exactly achieve what it aims at in the introduction, and overblows what it has achieved in the conclusion.

We have made significant attempts to present our findings in a more cautionary nature, especially with regards to conclusions drawn.

4. The figures are of very low quality.

We have made significant edits or removed all figures as originally submitted.

5. There are numerous mistakes in punctuation, and at least one in grammar. Please see attached pdf for detailed comments. I have not pointed out issue 5 in the comments.

We have made every attempt to eliminate all grammatical errors in the manuscript.

Comments from PDF:

Line 77 – I strongly recommend modifying the figures such that they will be genuinely informative for the reader not already familiar with the taxa--e.g. indicating where certain morphological features are, etc.

We agree and thank the reviewer for this helpful comment, we have made major changes to this figure and it now includes a more illustrative photograph of *Tribrachidium*, labels of relevant features as well as trace fossils associated with *Kimberella* and *Ikaria* and for the direction of movement in *Dickinsonia* so that it is more accessible to those unfamiliar with the fossil record of these organisms

Line 137 - non-metazoan holozoans--Holozoa refers to a clade containing Metazoa as well as Choanoflagellata, Ichthyosporea, and Filasterea

We have amended as recommended here and throughout. We also feel that addressing this issue in our outline of metazoan phylogeny at the beginning of the manuscript and in what is now Fig 1 helps to clarify this point.

Line 168 - There is a serious issue here:

AP patterning in bilaterians is associated with cWnt signalling because cWnt signalling is normally involved in gastrulation, which is what is also observed in Nematostella. However, Hox and ParaHox gene expression--associated with the AP axis of bilaterians--is in fact utilised in patterning the anthozoan directive axis, which is perpendicular to the oral-aboral axis specified by cWnt signalling. Therefore, it is unclear what the homology relations are between bilaterian and anthozoan body axes given this evidence alone, and thus inferring the patterning of the AP axis in Dickinsonia, Ikaria, and Kimberella by cWnt and Hox/ParaHox signalling is unwarranted, especially given uncertainty of axis homology between them and either bilaterians or cnidarians. See Nielsen, Brunet, and Arendt (2018) as well as Hoekzema et al (2017) for more information.

We thank the reviewer for bringing this inconsistency to our attention. Many of the issues highlighted are resolved by reference to “*perpendicular axes*” rather than A/P or D/V in Ediacara taxa as suggested by reviewer 2. Critically, none of these changes negate the major implications of this discussion, that axial patterning in representative Ediacara Biota taxa was likely controlled by Wnt, BMP, Hox and ParaHox signaling, as these are also used for axial patterning in a variety of metazoans. Discussion of this point is present in lines 223-228 and 320-331 including reference to Nielsen et al., 2018 and Hoekzema et al., 2017.

Line 179 - modify this sentence--without a clear definition of what dorsal and ventral mean (as you have done in the following sentences) this can be taken to mean that dorsal and ventral are defined here in terms of bilaterian dorsal and ventral sides, and homology relations between the body axes of these fossils and those of bilaterians are highly unclear.

Per reviewer 2 comments we have replaced initial assignment of A/P and D/V axes with “*perpendicular axes*” which resolves this issue.

Line 189 - This is an unwarranted over-generalisation: there are many White Sea taxa that aren't discussed here at all. please explicitly restrict your inference to the taxa discussed here.

We agree and this was not our intent. We have amended text here and elsewhere to more clearly state that our inferences are based on representative taxa and do not reflect the characters of all members of the Ediacara Biota

Line 195 - This is an unwarranted inference: it completely ignores the role of cBMP in specifying the directive axis of anthozoans--which casts into doubt its role in specifying the DV axis in the eumetazoan last common ancestor due to unclear homology relations between anthozoan and bilaterian axes--and thereby relies on the assumption that the Ediacaran AP and DV axes are homologous to bilaterian ones, which has not been established.

We agree with this statement and feel that description of “*perpendicular axes*” helps to resolve this issue. Importantly, as expressed by the reviewer, homology of these axes is not necessary in order to determine that BMP was likely involved in such patterning of representative Ediacara taxa.

Line 203 - sponges probably have a comparable degree of complexity with just a few cell types and still have epithelia, so why "although"?

This statement has been removed from the manuscript in response to reviewer comments.

Line 206 - regional differentiation doesn't straightforwardly suggest functional and therefore cellular differentiation

We agree and have significantly amended this discussion to lay out evidence for cell type and tissue differentiation in lines 202-221.

Line 218 - please improve the figure, especially (c). for a similar figure with higher quality see Dunn et al (2018)

Based on comments by the board member and all reviewers we have removed this figure.

Line 231 - "tagma" is the singular form of "tagmata"

We have amended to “*tagmata*” as recommended.

Line 253 - Absence of evidence is not evidence of absence: Notch/Delta may have been used in the development of features that would be invisible in Ediacaran fossils, such as germ cell differentiation.

We agree and have now explicitly state in lines 70-73 the assumption that because we are dealing with the fossil record we are reliant on an absence of evidence. We have also amended this specific statement to address the concern that Notch signalling was likely employed for other functions in lines 353-360.

Line 262 - its presence in a probably cnidarian says virtually nothing about its presence in the taxa under study here.

We agree and, based on comments by other reviewers, have removed discussion of *Haootia* from the manuscript.

Line 284 - again, non-metazoan holozoan

We have amended as recommended.

Line 288 - the independent origin isn't suggested by this evidence--it could easily have been present in the last common eumetazoan ancestor.

The independent origin of the nervous system in ctenophores, cnidarians and bilaterians is not a major finding of our work and further discussion is beyond the scope of this manuscript. Revised organization of the manuscript presents the inferred evolutionary progression of the nervous system at the beginning, which should address such confusion.

Line 295 - this only suggests the presence of mechanisms for signal transduction, not necessarily that of the nervous system. given that there is barely any evidence for the speed of such movements, it could well have been the case that these organisms used paracrine signalling, contractile epithelia with gap junctions (as is likely the case in placozoans), or hypothetical neuro-muscular cells.

Line 299 - this only suggests the presence of sensory cells, not neurons.

Line 306 - this doesn't actually follow: modern placozoans mainly use their outer rim for sensation, even though they use their entire ventral surface for feeding.

Line 308 - again, this is only about sensation and does not imply or suggest the presence of neurons--let alone their aggregation into nerve nets

Line 316 - this inference cannot be made based on the evidence presented: see all comments in this section

Given the similarity of these comments we will address them all in a single response. We appreciate the reviewers concerns regarding the sensory ability of placozoans and that this needs to be considered. We also agree that the presence of a nervous system in Ediacara taxa was not sufficiently argued in the original manuscript, however we contend that there is a significant body of evidence to suggest that these taxa possessed a rudimentary nervous system. Importantly, rapid and extensive movement in large Ediacaran taxa, sensory capabilities including at least two distinct sensory inputs, sediment displacement of *Ikaria*, and coordination between the proboscis and bottom “frill” adapted for mobility in *Kimberella* are consistent with the presence of a nervous system. We present this argument in lines 244-272.

Line 356 - it's the phenotype that achieves ecological success (or fails), not the mechanism underlying it

We agree and this description has been removed from the manuscript.

Line 362 - While this is a well-posed question and some evidence has been brought up to provide answers to it, the thing that does the actual explanatory work is not the genetic pathways--whose presence in extinct taxa cannot be safely inferred--but the morphogenetic processes that they underlie. Therefore, the loosely inferred presence of the genetic mechanisms aren't doing any explanatory work, especially given that their presence has been inferred via the inferred presence of the associated morphogenetic processes. Also bear in mind that morphogenetic processes do not stand in a one-to-one relation with genetic mechanisms, since genetic mechanisms often shift (as mentioned in the next paragraph), and thus precise mechanistic explanations are needed if the presence of one is to be inferred based on presence of the other.

We have rewritten much of the manuscript to address similar concerns, including this section. While we agree that morphogenetic processes are more confidently ascribed and contribute to explanatory power, we assert that identifying underlying genetic mechanisms is relevant and adds to our understanding of the Ediacara Biota. Increased caution has been added throughout the manuscript and we have laid out our assumptions (primarily of a metazoan affinity for representative taxa) initially to address concerns raised regarding potential issues of circularity.

Line 372 - it doesn't, and the significance of this claim is unclear anyway

We have removed this statement to address this concern.

Line 375 - this is a problematic line of argument. the reason is, again, that in order to infer the presence of the genetic mechanisms underlying morphological traits, it is the morphological traits and the morphogenetic processes underlying them that are being compared. Once these are compared, the sought-after phylogenetic information has already been obtained, and inferring the presence of certain genetic mechanisms, however satisfactorily argued, does not add any new information. This would not have been the case if we had direct access to developmental genetic evidence from the extinct taxa, which is obviously impossible to obtain.

We agree and have removed this statement to address this concern.

Line 401 - please specify what developmental capacity means here--it could refer to anything from certain genes to GRNs to morphogenetic processes

Line 404 - please briefly describe what Vendobionta refers to

This section has been removed, as much of the discussion is unnecessary given the assumption of metazoan affinities for representative taxa.

Line 414 - I don't see why this is surprising, given that our best current evidence suggests that the earliest bilaterians were likely relatively simple organisms and acquired complex traits later independently

Molecular clock evidence supports the divergence of bilaterian phyla in the Ediacaran (e.g. dos Reis et al., 2015) that might be expected to exhibit these characters, however, we agree that the potential simplicity of the PDA may partially resolve this issue and should be acknowledged, see lines 416-417.

Line 425 - consider improving the figure

We have revised this figure (now Fig 1) and feel it more appropriately represents the relationships between representative taxa and modern relatives based on developmental characters and underlying regulatory control.

Line 430 - even granting the presence of these signalling pathways, there is no reason to assume that they were co-opted rather than ancestrally conserved in their roles.

We agree that these most likely represent ancestrally conserved roles and this sentence was only meant to add some caution to our interpretations. We have removed the statement as the necessary caution has been stated elsewhere in the manuscript in response to comments by all reviewers.

Line 438 - please spell out what these stand for

We have spelled out topologically associated domains (TADs) and more broadly referred to their associated insulator proteins.

Line 449 - once again, crucial evidence from *Nematostella* is being ignored.

We agree and have amended this discussion to instead refer to “*axial polarity*” and “*eumetazoan organization*” to address this concern.

Line 459 - I take this to mean independently in various bilaterian lineages. In this case, the absence of heads in Ediacarans is merely consistent with, and doesn't actually support, independent evolution of the CNS in bilaterians because it could also be the case that the last common bilaterian ancestor had a CNS which was lost by likely bilaterian Ediacarans such as *Kimberella*.

We have amended this statement to indicate consistency rather than support. With respect to the possibility that the lack of a CNS could represent independent loss, we acknowledge that this is possible, but given that these are the earliest known bilaterians in the fossil record and such features are absent in a variety of such forms, we consider this unlikely. This would also be inconsistent with results based on modern regulatory controls of these structures which, as stated in the manuscript, suggest the independent evolution of these characters in disparate bilaterian phyla. We have addressed the possibility of loss in lines 438-441.

Line 470 - "reveals" is a strong term, especially given the absence of direct evidence--the evidence presented here at best "suggests" their presence.

We have amended as suggested.

Line 470 - A similarly strong word used a few times in this paper is "implies". I recommend changing them too.

We have replaced all instances of the term “implies” in this manuscript.

Line 471 - "pointed out" rather than "identified"

We have amended this statement so that it more appropriately refers to the “*identification*” of morphogenetic processes rather than likely regulatory elements.

Line 472 - this isn't what you've done--you've taken metazoan affinities of the Ediacaran organisms under discussion as an assumption which has been necessary for the inferences made throughout the paper.

We agree with this statement and now take the metazoan affinities of these taxa as an initial assumption and removed this statement from the conclusion (although see discussion of the similarities between morphogenetic processes in modern metazoans and representative Ediacara taxa in lines 373-393).

Line 476 - what this analysis actually supports is that these organisms potentially fill the gap in the evolution of the morphological features described here (e.g. body polarisation followed by nervous system development). All inferences about the genetic mechanisms underlying these features are secondary and, as discussed in previous comments, don't really add anything.

The pattern identified in the first sentence of this comment is precisely what we envision as the significance of this contribution. We have now made a stronger, yet cautious argument for the presence of a nervous system and so feel this is warranted as a likely feature of the Ediacara Biota. We have also amended this statement to refer more broadly to the characters rather than their underlying regulatory control.

Line 477 - since this is not an empirical paper, use of the word "results" doesn't make much sense here. the presence of bilaterians in the Ediacaran is taken from molecular clock studies and the placement of *Kimberella* and *Ikaria* within Bilateria in other studies, and is not an original contribution made here.

We acknowledge that this is not a traditional research article and have removed the term “results” from this statement and elsewhere in the manuscript. While we agree that other lines of evidence previously published support the placement of *Kimberella* and *Ikaria* within Bilateria but maintain that the observations made here contribute to our confidence in that classification.

Line 478 - again, this is not a contribution of this study.

We feel that our analysis does lend support to a growing body of evidence that the Ediacaran included bilaterians and the genetic mechanisms critical to their later success. A more cautionary description of the work and removal of the phrase “our results” from this statement are attempts to address this concern.

Reviewer 2

The article by Evans et al “Developmental processes in Ediacaran macrofossils” is an interesting idea to use developmental characters to place ediacarians in the Metazoan phylogeny and also to characterize the fossilized. It is NOT a research paper, as it has been categorized, it is a review or perspective. So it should be declared as this, since there is no new data provided.

We are happy to have this contribution classified as a review but do contend that our approach results in the novel classification of a suite of likely regulatory elements responsible for representative Ediacara body plans and critical developmental features such as the presence of a nervous system but lack of centralization.

I find this approach also a bit circular, reminding me on the old days where animal phylogenies were purely based on morphological characters. However, since we deal with fossils here and the animal phylogeny provides a relatively solid frame, it is ok to proceed like this.

We agree and have made extensive revisions to avoid the circular reasoning apparent in the original version of this manuscript. We also agree that though sometimes unsatisfactory we are limited by the information presented in fossil record. This has been stated explicitly in lines 70-73.

The whole paper lacks at some parts the necessary accuracy in the description of the molecular components. The authors need to rework the wording and make it more specific and expand the categories, such as structural proteins (which is NOT a pathway), transcription factors and signalling cascades in Table 1.

We have made every attempt to present a complete description of all molecular components within the scope of a single manuscript. We have also amended Table 1 to refer to “*Regulatory control*” which should address any mechanism responsible for the characters identified.

Furthermore, I find the references a bit randomly chosen and some of the key papers are not listed (e.g. the Hox in *Nematostella* DOI: 10.1126/science.aar8384, the actin-myosin presence and function in choanoflagellates has been shown much earlier than Brunet et al). The manuscript would gain by adding more references and reworking the present.

We have expanded references throughout the manuscript based on these and similar comments, including reference to He et al., 2018 and edited reference to actomyosin based contractility to include previous research that has demonstrated their presence in choanoflagellates.

I am also not friend with the Figures. Figure 1 is ok and necessary, but I don't know the function of Figure 2 and I do not think it is necessary at all.

We agree and have made improvements to Fig 1 (now Fig 2) and removed our original Fig 2 to address these and other reviewer concerns.

Figure 3 has a very ladder-like impression, picturing the great chain of beings and must be reworked. It also clearly can be made more attractive.

We agree and have revised our original phylogeny (now Fig 1) from a “ladder-like” schematic to a tree diagram, which we feel more appropriately represents the relationships between representative taxa and modern relatives based on developmental characters and underlying regulatory control. We hope the reviewer will find improved line drawings more attractive.

See below some remarks to the characters:

Multicellularity

Coming with the assumption that the ediacarians are Metazoa, it is obvious to assume their multicellularity.

We agree that multicellularity is obvious given the assumption of metazoan affinity. However, we feel that this character and regulatory elements obviously present is a useful starting place for the line of reasoning presented throughout, and so have left it in the manuscript.

The authors should maybe include the hallmark and apomorphy of Metazoa, which is the sperm and oocyte production through meiosis. Also means the germ line, likely specified through PIWI etc.

With respect to sperm and oocyte production through meiosis, we have elected not to include this, especially given the comment by reviewer 3 that inferences of reproductive mode are highly speculative.

Anterior-Posterior and Dorsal-Ventral polarity

I suggest the authors change it to perpendicular body axes. It can still be related to the same molecules, but even in Cnidaria it is not clear what axis can be related to the Bilateria. E.g. the BMP-chordin axis is the same as the Hox-axis and even connected through regulation (see <http://dx.doi.org/10.1016/j.celrep.2015.02.035>). The Wnt axis in cnidarians is perpendicular to this. Therefore it is to assume that the axial patterning systems were present, but they can not be related to A-P or D-V. So the authors should change this and it also makes the whole section of guessing what is D-V or A-P in these animals obsolete.

We agree and thank the reviewer for this constructive suggestion. We now refer to “*perpendicular axes*” or “*axial polarity*” when using these to infer potential underlying genetic control.

Differentiation

- The authors state that “coeloms” are present possibly in Kimberella and Ikaria (line 208) and connect this to the establishment of three germ layers. However, it is not necessarily the case. there are many kinds of cavities in animals that can be formed without the presence of three germ layers (gastric pouches in cnidaria, other cavities in ctenophores, pseudocoeloms etc.). So it is not necessarily the case. Here, in this work, the authors aim and should aim for the minimal presence of characters and try to avoid overestimations.

We agree with this statement and have removed discussion of a potential coelom.

The role of brachyury is not clear. What is clear is, that it has many functions in cell internalization (not cell migration), not necessarily connected to mesoderm. I suggest to eliminate brachyury from the whole paper, because its pure speculation.

We thank the reviewer for this comment and have removed all reference to *brachyury*.

Line 240: Hox are not “establishing” polarity, but pattern along the axis. Polarity is established much earlier, namely by Wnt in most cases.

We agree and have amended to address this concern.

Nervous System:

In general I feel that Placozoans are not considered sufficiently in the manuscript. It is to assume that placozoans can sense their environment without nervous system. I think they play an important role in the section of the nervous system as do carnivorous sponges for example.

We agree and thank the reviewer for highlighting this shortcoming. We have added discussion of placozoans and sponges as they relate to the development of a nervous system, lines 247-250 and 264-268.

Referee: 3

Comments to the Author(s)

Evans and colleagues characterise key features of a select suite of fossil organisms from the Ediacaran macrobiota, aiming to identify key anatomical features that may be linked to molecular genetic developmental processes of animal bodyplan organisation - which they review. Their explicit aims are to:

"First, by integrating insights from the fossil record with comparative data from living taxa, can reliable inferences be made about the developmental capacity of Ediacaran clades?"

"Second, can we infer plausible phylogenetic positions for and/or relationships between these taxa with the addition of this developmental information?"

They conclude that the considered elements of the White Sea macrobiota provide evidence for "several metazoan specific regulatory elements, suggesting that the Ediacara Biota included animals (e.g. [1, 11]). Our analysis further supports that many of these organisms occupy the significant gap between the adaptation of signalling pathways responsible for body polarization and nervous system development and their subsequent co-option for more specific regionalization and the formation of a CNS."

I am extremely sympathetic to the aims of this study, however, I am concerned that, as presented, it lacks a logical basis for inference. Specifically, the search for evidence of metazoan

developmental mechanisms assumes that these organisms are animals and, yet, this is presented as a conclusion. The fossils are used to infer the sequential acquisition of developmental mechanisms as well as the phylogenetic affinity of the organisms based on the anatomical characteristics, rooted in developmental mechanisms, that the fossils exhibit; it is not possible to do the former without the latter being established a priori.

In summary, the logic presented in the manuscript reads as circular. Much of the text is review rather results and it is not clear that the review yields any added value. I would suggest that the manuscript/study is revised with a more logical inference framework:

1. Set out the phylogenetic affinity of living clades first
2. Infer anatomical character evolution within this framework
3. Review regulatory evolution within this framework
4. Resolve affinity of fossils in this framework
5. Evaluate what impact fossils have on inferences of anatomical character and regulatory evolution

This structure includes many of the same components of the current manuscript but, crucially, the process of inference is linear; assumptions and insights are more readily distinguishable.

We thank all reviewers for comments regarding the apparent circularity of our arguments as previously constructed. We have now adopted the proposed structure outlined above. We believe this more convincingly argues the points of the original manuscript.

At present, the absence of an explicit phylogenetic hypothesis of living organisms is a major limitation. How are the described clades defined? Does Bilateria equate to Nephrozoa or does it also encompass Xen/aceolomorpha? Where do the authors resolve ctenophores and, if they consider the position of ctenophores uncertain, what impact does this have on their analysis?

We agree and have now included a section discussing our inferred metazoan framework as part of the Introduction as well as revised our phylogeny to include this framework in what is now Fig 1.

Crucially, what are the characters that supports the assignment of the fossils to specific positions within the phylogenetic tree?

We agree that the previous version of this manuscript was not explicit in the characters used to differentiate taxa. This figure (now Fig 1) and the text (lines 395-410) describing our arguments for the phylogenetic positions of these taxa has been edited to address this concern.

Details

Lines 51-53: Many taxa from the Ediacara Biota may represent stem lineages of major animal groups but their diagnostic characters either were not preserved or had not yet evolved.

Be specific WRT to 'stem lineages of major animal groups' since Metazoa and Eumetazoa and Bilateria are major animal groups, the diagnostic characters of which are preserved, hence this

study. I think, rather, that the authors are referring to phyla and their constituent clades.

We have amended as suggested, replacing “groups” with “*phyla and their constituent clades*”.

Lines 147-152: Given similarities with animals, protein domains essential for multicellularity found in holozoans and basal metazoan clades were likely present in Ediacaran taxa. For example, actomyosin-based contractility of cell sheets (necessary for epithelia formation) has been described in choanoflagellates [35] and ichthysporeans [70]. Regulatory elements likely included multiple extracellular matrix domains and transcription factor families, such as cadherins, C-type lectins, tyrosine kinases, LIM Homeobox and canonical Type IV collagens, among others [36-38].

Do you mean protein domains (the structural modular components of proteins) or protein families? Why are these likely present in Ediacaran taxa? Unless you assume that these fossil organisms are crown holozoans? In which case, what is the insight?

We agree that beginning with the assumption of metazoan affinity not much is gained from the description of representative taxa as multicellular. However, we feel this represents a character that has logical grounding in observed morphologies (based on their size, differentiated macroscopic structures, etc) and can be tied to specific regulatory elements. The utility of including this is as an example of the line of reasoning presented throughout.

Lines 157-159: Although polarization in Dickinsonia may not be homologous to the A/P axis in bilaterians [11], it possessed the developmental capacity to produce an A/P axis and differentiate between two “ends” relative to the direction of movement [26].

How does this follow? Why not another axis?

We agree that this statement was confusing. Reference to A/P polarity in this manner has been amended to refer instead to “*perpendicular axes*” or “*axial polarity*” as suggested by reviewer 2.

Lines 180-182: We consider the surface preserved facing upwards into the overlying sand as the top or dorsal surface and the opposite as bottom or ventral. As with A/P polarity, it is the ability to differentiate these that is important here.

What are we learning here? Affinity or phylogenetic position of the fossil? When the mechanism evolved?

This has been amended to address similar concerns by other reviewers. The significance here is that we can identify the ability to specify perpendicular axes indicative of a suite of regulatory mechanisms responsible for their formation.

Lines 199-214: Differentiation

The scale and patterning of numerous Ediacara Biota taxa implies multiple cell types and some degree of regional differentiation. Early forms, such as fractally organized rangeomorphs, may have been possible with a few cell types, although a recent reconstruction suggests at least a differentiated epithelium (Butterfield, 2020). Differentiation of White Sea taxa, such as

Dickinsonia, is less clear, although an undivided anterior region (Fig 2a) and possible musculature [24, 26, 27] (see below) support some level of cellular disparity. Kimberella has perhaps the strongest evidence for distinct cell-types and regional specification of functional units (Fig 2b). The ability to excavate the organic mat and displace sediment grains in Kimberella and Ikaria supports the presence of a coelom [21, 74]. This is unique to bilaterians and signifies the establishment of three distinct germ layers. Further, feeding in Kimberella [17, 18] and Ikaria [21, 22] suggests that they had a mouth and some type of gut, potentially a through-gut, although direct morphological evidence for these structures has yet to be recognized. While coeloms evolved several times among bilaterians, the presence of coelom and gut required the ability to form multiple tissue layers.

Cell types or tissues or organs? Epithelia only require cell polarization, not necessarily differentiation of cell types. Organs require epithelia. Focus on what you are inferring.

This point is well taken, and we have now expanded this discussion to address this concern, namely specifying that we infer distinct cell-types and tissues, but cannot confidently identify organs, with the possible exception of *Kimberella proboscis* and/or muscular foot, lines 202-221.

Musculature: locomotion. Why stray into other taxa like Haootia? Are you considering them together because you think they are a clade and provide reciprocal insights?

We agree and have removed reference to *Haootia*.

CNS complexity: evidence of absence or absence of evidence?

We have amended the text to explicitly state that, because we are reliant on the fossil record, we can only demonstrate an absence of evidence, lines 70-73. However, given extensive evidence of such features in the Cambrian fossil record we feel this absence is potentially meaningful. See discussion in lines 275-278.

Problematic characters: sexual reproduction rests too much on inference from size classes

Discussion of reproductive modes has been removed to address this concern.

Discussion:

Inferences of characters have already assumed affinity, to some extent at least, and so you cannot use these inferences to infer phylogenetic position, either of the fossils or the timing of origin of regulatory factors.

This discussion does not build on what goes before; does not formally interpret the 'results'.

The new organization proposed by this reviewer addresses many of these concerns. As this may be better classified as a review the heading "discussion" has been removed. We do assert that beginning with the assumption of an animal classification we can gain insight into the placement of representative taxa within Metazoa and possible regulatory pathways.